# *Lactiplantibacillus (Lactobacillus) plantarum* as a Complementary Treatment to Improve Symptomatology in Neurodegenerative Disease: A Systematic Review of Open Access Literature

**DOI:** 10.3390/ijms25053010

**Published:** 2024-03-05

**Authors:** Ana Isabel Beltrán-Velasco, Manuel Reiriz, Sara Uceda, Víctor Echeverry-Alzate

**Affiliations:** Psychology Department, School of Life and Nature Sciences, Nebrija University, 28240 Madrid, Spain; abeltranv@nebrija.es (A.I.B.-V.); mreiriz@nebrija.es (M.R.)

**Keywords:** neurodegenerative disease, probiotics, Alzheimer’s disease, Parkinson’s disease, Multiple Sclerosis, neurodegenerative process, *Lactiplantibacillus plantarum*, *Lactobacillus plantarum*

## Abstract

This systematic review addresses the use of *Lactiplantibacillus* (*Lactobacillus) plantarum* in the symptomatological intervention of neurodegenerative disease. The existence of gut microbiota dysbiosis has been associated with systemic inflammatory processes present in neurodegenerative disease, creating the opportunity for new treatment strategies. This involves modifying the strains that constitute the gut microbiota to enhance synaptic function through the gut–brain axis. Recent studies have evaluated the beneficial effects of the use of *Lactiplantibacillus plantarum* on motor and cognitive symptomatology, alone or in combination. This systematic review includes 20 research articles (*n* = 3 in human and *n* = 17 in animal models). The main result of this research was that the use of *Lactiplantibacillus plantarum* alone or in combination produced improvements in symptomatology related to neurodegenerative disease. However, one of the studies included reported negative effects after the administration of *Lactiplantibacillus plantarum*. This systematic review provides current and relevant information about the use of this probiotic in pathologies that present neurodegenerative processes such as Alzheimer’s disease, Parkinson’s disease and Multiple Sclerosis.

## 1. Introduction

### 1.1. The Neurodegenerative Disease

Neurodegenerative diseases refer to a group of pathologies that mainly affect the Nervous System (NS), and which involve tissue degeneration and cell death [1,2]. Lesions can be located in any region of the NS (spinal cord, brain, nerves, etc.), and affect cognitive and/or motor functioning. The impact of these pathologies is very significant [3,4]. For example, it is estimated that in 2050, 153 million people will have dementia [5]. The prevalence of Alzheimer’s disease is particularly relevant and currently affects almost 47 million people worldwide. Regarding Parkinson’s disease, 8.5 million people currently suffer from it [5,6].

One characteristic that is usually present in neurodegenerative diseases is the damage associated with oxidative stress [7,8,9,10]. Thus, it is traditionally studied in relation to cognitive impairment and neurodegeneration processes [11,12]. This phenomenon occurs when there is an imbalance between pro-oxidant and antioxidant chemicals. These pro-oxidants contain free electrons (which are also called free radicals) that have the ability to produce oxidative damage in fractions of a millisecond. However, this damage is sufficient to cause apoptosis and necrosis [13,14]. These mechanisms of action are involved in a number of degenerative pathologies, although it is not possible to confirm exactly whether they are the cause or effect of the clinical disease [15,16,17].

More than 100 neurodegenerative diseases or conditions associated with neuronal death have been described, among which Alzheimer’s Disease (AD), Parkinson’s Disease (PD) and Multiple Sclerosis (MS) stand out due to their prevalence [18,19]. AD is the most common form of dementia among older adults, with a prevalence of more than 46 million people affected worldwide [20,21].

Regarding PD, it is a neurodegenerative disease with premature death of dopaminergic neurons in the substantia nigra pars compacta in the midbrain [22,23]. In addition, atrophy of the basal ganglia is present, as well as dysfunction of communication between these subcortical regions and the prefrontal cortex [24,25,26,27]. Common symptoms in this disease are parkinsonism (bradykinesia and tremor or rigidity), postural instability and gait disturbances [28,29]. In terms of non-motor symptoms, these patients frequently present neuropsychiatric disorders, sleep disorders, autonomic dysfunction and gastrointestinal symptoms, among others [30,31,32].

Finally, MS is an autoimmune disease, although it causes degeneration in the brain and spinal cord. The damage is caused by an inflammatory process, which occurs when the immune system’s own cells attack the NS [33,34]. Because the lesions can be located in different regions, the symptoms are heterogeneous [35,36,37]. The same happens with each flare-up, when new lesions occur anywhere in the brain, optic nerve or spinal cord [38]. Among the most frequent symptoms, we find intestinal symptoms, such as urinary urgency or incontinence; muscular symptoms, such as spasms, stiffness, difficulty in coordination and gait; ocular symptoms such as discomfort, nystagmus or loss of vision; neurological symptoms such as impairment of executive functions such as memory or attention and difficulty in problem solving, among others [39,40,41,42].

### 1.2. The Impact of Gut Microbiota in Neurodegenerative Disease

There is growing evidence linking the gut microbiota (GM) with neurodegenerative diseases, but it is unclear if it is the consequence or the cause of the disease. The GM is formed by more than 100 million bacteria and more than 300 different species which live in our digestive tract [43,44,45,46]. Bacteria belonging to the phylum *Bacillota* (*Eubacterium, Dorea, Ruminococcus*), the phylum *Actinomycetota* (*Bifidobacterium*) and the phylum *Bacteroidota* (*Alistipes, Bacteroides*) are essential in the gut microbiota [47,48,49]. In addition, the bacteria, the GM is comprised of other microorganisms such as archaea, viruses, fungi, and protozoa [50,51].

It is known that the GM metabolites are involved in the onset and/or evolution of neurodegenerative diseases through different action mechanisms that alter endocrine pathways, immunological signals or neurological factors, among others [52,53]. When the GM is preserved, we talk about eubiosis; however, in some patients there is an alteration of the ratios of different species, and in this case, we speak of dysbiosis [54,55,56,57]. GM dysbiosis has been associated with neurodegenerative processes related to pathologies such as PD, AD or MS [58,59], and is usually accompanied by an increase in *Bacteroidota* as well as a reduction in the levels of *Bacillota* and *Bifidobacterium* [60,61,62].

This field of study is currently seen as an opportunity for intervention in these patients. The results obtained with the use of probiotics (live microorganisms that can be orally administrated as a food supplement or medicine) indicate that it is possible to modulate the microbiota, reducing organic inflammation and the symptoms associated with these neurodegenerative diseases [63,64,65]. Finally, the gut microbiota can modify the expression of certain genes that have been associated with different neurodegenerative diseases, as well as those involved in the mechanisms of cognitive functioning [66,67]. In addition, GM modulates metabolic functions, downregulating the low-grade inflammatory processes [68,69].

Among the most studied probiotics in different pathologies are *Bifidobacterium* (mainly *longum* and *bifidum* spp.) and the *Lactobacillaceae* family (mainly *Lactobacillus paracasei*, *Lactobacillus casei* and *Lactoplantibacillus plantarum* spp.) [70,71,72]. *Bifidobacterium* is a genus of gram-positive, anaerobic bacilli, and they utilize the fructose-6-phosphate phosphoketolase pathway. *Lactobacillaceae* is a family of long, curved or straight, gram-positive, anaerobic, lactic acid-forming bacilli [73,74,75,76,77]. Of these, *Lactiplantibacillus (Lactobacillus) plantarum* appears to have the most promising results in the intervention of symptoms associated with neurodegeneration [78,79,80]. Several authors have suggested that the use of *Lactiplantibacillus plantarum*, alone or in combination, improves specific symptoms such as motor, cognitive and psychiatric signs in neurodegenerative processes [81,82,83,84,85,86,87,88,89]. Research regarding these effects has increased in the last decades, making it necessary to compare with the results obtained. It is important to note that the term *Lactobacillus plantarum* was modified in 2020 for *Lactiplantibacillus plantarum*. Thus, in this Systematic Review, studies that included both terms were registered to ensure that all potential studies have been included.

The main objective of this Systematic Review was to collect all significant findings from both human and animal studies investigating the impact of *Lactiplantibacillus plantarum* administration on the improvement of motor and/or cognitive symptomatology in patients with neurodegenerative disease. This review will provide the most current data on the use of *Lactiplantibacillus plantarum* and intends to facilitate future research in this area.

## 2. Methods

### 2.1. Literature Search

The review followed the Preferred Reporting Items for Systematic Reviews and Meta-Analyses (PRISMA) statement [90,91]. We searched for articles published between 2000 and 2024 in the following databases: ScienceDirect, Scopus and Web of Science. We used the following search: *(Lactobacillus plantarum OR Lactiplantibacillus plantarum)* AND *Neurodegenerative disease*. Two authors independently conducted the literature search in January 2024 (A.I.B.-V and S.U.), including the initial review of titles and abstracts, and the evaluation of retrievable articles for comprehensive review. Included articles were original research studies in English and Open Access.

### 2.2. Study Selection

In both human and animal studies, the inclusion criteria required: (i) a direct relationship between the use of *Lactobacillus plantarum* OR *Lactiplantibacillus plantarum* and changes in the symptomatology of neurodegenerative disease; (ii) the use of *Lactobacillus plantarum* OR *Lactiplantibacillus plantarum* as a therapeutic target in the process of neurodegeneration was addressed; (iii) the studies included any of the pathologies studied in this systematic review, which involve neuronal death associated with AD, PD, MCI and MS, oxidative stress and other neurodegenerative processes.

## 3. Results

A total of 907 articles were retrieved (Figure 1), of which only 29 articles were identified that fulfilled the eligibility criteria. Once duplicates were removed, the full titles and abstracts of all articles were examined for eligibility. Studies that addressed *Lactobacillus plantarum* OR *Lactiplantibacillus plantarum* unrelated to the neurodegeneration process were excluded: 5 were studies that did not include intervention with *Lactobacillus plantarum* OR *Lactiplantibacillus plantarum* and 4 were studies that focused on other pathologies not studied. Following this detailed review, a total of 20 studies were included. After the screening phase, all the selected articles were retrieved for comprehensive review, based on the established inclusion criteria.

### 3.1. Data Extraction

The data extracted from the included studies were as described below (Table 1):▪Type of study (human or animal models);▪Type of probiotic used (*Lactobacillus plantarum* or *Lactiplantibacillus plantarum* alone or in combination);▪Neurodegenerative pathology addressed in the research;▪Population (description);▪Methodology of the research carried out;▪Intervention (dose administered, time);▪Results obtained after the intervention with Lactobacillus plantarum or Lactiplantibacillus plantarum.

### 3.2. Human Model

Three studies were conducted in humans: two address PD and the impact of the use of *Lactobacillus plantarum* or *Lactiplantibacillus plantarum* in the intervention of neurodegenerative processes in PD patients [92,93] (*n* = 2), and 1 of them explored MCI (Mild Cognitive Impairment) [94] (*n* = 1).

#### 3.2.1. Parkinson’s Disease

Ghyselinck et al. (2021) used stool samples from three PD patients to create an in vitro intestinal model and dosed Symprove, a probiotic consisting of *Lactobacillus acidophilus* NCIMB 30175, *Lactobacillus plantarum* NCIMB 30173, *Lactobacillus rhamnosus* NCIMB 30174 and *Enterococcus faecium* NCIMB 30176 [92]. After 48 h of probiotic dose, the microbial community analysis showed significant changes, with elevated levels of *Firmicutes*, *Actinobacteria* and *Bacteroidetes*. In addition, the production of short-chain fatty acids (SCFA) and lactate was stimulated. Levels of anti-inflammatory cytokines (IL-6, IL-10) were also increased and levels of pro-inflammatory cytokines and chemokines (MCP-1, IL-8) were decreased (TEER-transepithelial electrical resistance (110.3 ± 1.3, *p* < 0.001)).

Another study conducted by Lu et al. (2021) analyzed the use of *Lactobacillus plantarum* PS128 for 12 weeks. The results showed that after 12 weeks of PS128 supplementation, UPDRS (Unified Parkinson’s Disease Rating Scale) motor scores improved significantly in both the OFF (−0.80 ± 1.85, *p* = 0.04) and ON (−2.56 ± 5.36, *p* = 0.007). In addition, PS128 intervention significantly improved ON and OFF period duration as well as PDQ-39 (Parkinson’s Disease Questionnaire) values (−5.68 ± 8.55, *p* = 0.031). However, no apparent effect of PS128 on non-motor symptoms of PD patients was observed [93].

It can be concluded that *Lactiplantibacillus* (*Lactobacillus*) *plantarum* improves motor symptomatology in PD due to its neuroprotective action through its microbial profile [92]. Although a significant effect on non-motor symptomatology could not be established, microbial modulation allows for the modification of the protective effects of anti-inflammatory cytokines [93].

#### 3.2.2. Mild Cognitive Impairment

Fei et al. (2023) conducted a study with 42 patients diagnosed with ICM to whom they administered a probiotic mixture (*Lactobacillus plantarum* BioF-228, *Lactococcus lactis* BioF-224, *Bifidobacterium lactis* CP-9, *Lactobacillus rhamnosus* Bv-77, *Lactobacillus johnsonii* MH-68, *Lactobacillus paracasei* MP137, *Lactobacillus salivarius* AP-32, *Lactobacillus acidophilus* TYCA06, *Lactococcus lactis* LY-66, *Bifidobacterium lactis* HNO19, *Lactobacillus rhamnosus* HNO01, *Lactobacillus paracasei* GL-156, *Bifidobacterium animalis* BB-115, *Lactobacillus casei* CS-773, *Lactobacillus reuteri* TSR332, *Lactobacillus fermentum* TSF331, *Bifidobacterium infantis* BLI-02 and *Lactobacillus plantarum* CN2018), for 12 weeks. The results showed that the probiotic mixture improved cognitive function as well as sleep quality (Mini-Mental State Examination-MMSE (24.75 ± 2.47); Montreal Cognitive Assessment Scale-MoCA (22.05 ± 2.14 vs. a 20.10 ± 1.45); Pittsburgh Sleep Quality Index-PSQI (5.35 ± 2.78 vs. 8.40 ± 1.76,) (* *p* < 0.001).

In conclusion, the evidence suggests that treatment with this mixture of bacterial strains, which includes different species of *Lactobacillaceae*, can improve mild cognitive impairment and sleep quality [94].

### 3.3. Animal Model

A total of 17 studies were conducted using different animal models. Of these, 4 focused on AD [95,96,97,98] (*n* = 4); 7 focused on PD [99,100,101,102,103,104,105] (*n* = 7); 1 analyzed MS [106], (*n* = 1); 3 studied oxidative stress [107,108,109] (*n* = 3); 1 on cognitive impairment and oxidative stress [110] (*n* = 1); and 1 on neurodegenerative process [111] (*n* = 1). All of them addressed the intervention with *Lactobacillus plantarum* or *Lactiplantibacillus plantarum* for the improvement of symptoms associated with neurodegenerative pathologies.

#### 3.3.1. Alzheimer’s Disease

Bonfili et al. (2020) conducted a study with a transgenic mouse model and administered the probiotic SLAB51 for a period of 16 to 48 weeks [95]. It contained eight different live bacterial strains: *Streptococcus thermophilus* DSM 32245, *Bifidobacterium lactis* DSM 32246, *Bifidobacterium lactis* DSM 32247, *Lactobacillus acidophilus* DSM 32241, *Lactobacillus helveticus* DSM 32242, *Lactobacillus paracasei* DSM 32243, *Lactobacillus plantarum* DSM 32244 and *Lactobacillus brevis* DSM 27961. The results obtained indicate that the ingestion of probiotics leads to an improvement in the altered brain glucose metabolism associated with the pathogenesis of AD, thereby delaying the progression of the disease (SLAB51 (* *p* < 0.05) vs. control and Veh.).

The study conducted by Lee et al. in 2021 in a transgenic mouse model addressed the impact on AD symptomatology of a probiotic composed by *Lactobacillus plantarum* NK151; NK173 and *Bifidobacterium longum* NK173 [96]. The results showed that the intervention reduced memory impairment and modulated gut bacterial composition in mice with AD cognitive impairment.

Mallikarjuna, Praveen, and Yellamma (2016) conducted a study in D-galactose-induced AD rats, and administered *Lactobacillus plantarum* MTCC1325 for 60 days [97]. The results demonstrated that, within a 30-day period, rats in the probiotic group exhibited a reversal of all ATPase enzyme components to normal levels. These data indicates that these probiotics exerts a protective action on the ATPase system in the brain, reducing the neurodegeneration (Alleviated *Escherichia coli* K1-induced cognitive impairment (# *p* < 0.05 vs. NC. * *p* < 0.05 vs. EC.).

A study conducted by Kaur et al. in 2021 in an AppNL-G-F (AD) mouse model analyzed the impact of the probiotic compound VSL#3^®^ (*Lactobacillus plantarum, Lactobacillus delbrueckii subsp. Bulgaricus, Lactobacillus paracasei, Lactobacillus acidophilus, Bifidobacterium breve, Bifidobacterium longum, Bifidobacterium infantis* and *Streptococcus salivarius* subsp. *Thermophilus*) for 8 weeks [98]. This research was focused on finding differences by gender, and the results indicated that female mice had improved mnesic function, although no improvement was found in male mice. Similarly, a reduction in TNF-α levels in the brain was found in female mice (*Significant difference at *p* < 0.05 vs. control group. #Significant difference at *p* < 0.05 vs. AD group).

These results suggest that *Lactiplantibacillus plantarum* improves mnesic function, acts as a neuroprotectant in hippocampal cells delaying the neurodegenerative process [95,96], and reverses the action of ATPases in the brain [97]. Moreover, the study that focused on gender differences underscores the importance of conducting studies to analyze potential differences in the effects of probiotic use based on gender and to comprehend the mechanisms involved in these effects [98].

#### 3.3.2. Parkinson’s Disease

Liao et al. (2020) conducted an investigation with a mouse model of MPTP-induced PD [99]. They administered *Lactobacillus plantarum* PS128 for 28 days and an increase in DA and norepinephrine were found in the striata. In addition, *Lactobacillus plantarum* PS128 was shown to reduce the death of nigrostriatal dopaminergic neurons. Similarly, *Lactobacillus plantarum* PS128 reduced MPTP-induced glial reactivity, and increased striatal neurotrophic factors (PS128 increased the DA levels and significantly improved the DOPAC (*p* < 0.05), HVA (*p* < 0.001), NE (*p* < 0.01) and MHPG (*p* < 0.001) levels in the MPTP-treated group).

Lee et al. (2023) conducted a study in a mouse model of PD [100]. *Lactobacillus plantarum* PS128 was administered for 6 weeks and the results showed a significant reduction in motor deficits and higher dopamine levels (PS128 significantly increased DA levels vs. ROT group (*p* < 0.05)). Similarly, a reduction in the loss of dopaminergic neurons and microglial activation was found. Levels of inflammatory factors were also reduced and an increase in the expression of neurotrophic factor was shown in the brain. Ingestion of *Lactobacillus plantarum* PS128 modified the microbial profile of PD mice and sustained neuroprotective effects by increasing the expression of the suppressor cytokine signaling 1 (SOCS1).

Along the same lines, Wang et al. in their 2021 study in a mouse model of PD induced by MPTP (1-methyl-4-phenyl-1,2,3,6-tetrahydropyridine) [101], administered *Lactobacillus plantarum* DP189 for 14 days. The results indicated that this product exerted a significant neuroprotective effect on dopaminergic neurons, which improved motor symptoms (activities increased by 13.8% vs. model group (*p* < 0.05)). In addition, an increase in the levels of monoamines in the nervous system, namely 5-HT (5-hydroxytryptamine) and DA (dopamine) was also described. Another interesting result was that *Lactobacillus plantarum* DP189 promoted neuronal survival through a modulation of the ERK2 and AKT/mTOR pathways.

Pérez Visñuk, et al. conducted a study in 2020 [102], in a mouse model of PD induced by 1-methyl-4-phenyl-1,2,3,6-tetrahydropyridine. They administered *Lactobacillus plantarum* CRL 2130, *Streptococcus thermophilus* CRL 807 and *Streptococcus thermophilus* CRL 808 alone or in combination. The results revealed a notable enhancement in the motor skills of mice under both individual and combined administrations; however, the combined form appeared to be more effective. Moreover, the mechanisms of action did not appear to be entirely clear.

Pérez Visñuk et al. in 2022 conducted another study in a mouse model with PD induced by MPTP and probenecid, to whom they administered *Lactobacillus plantarum* CRL2130 [103]. The results showed that motor deficits were reduced, in addition to preventing the death of dopaminergic neurons. A reduction in proinflammatory cytokines and an increase in IL-10 were also observed compared to the control group, showing a neuroprotective effect. In this same study an in vitro analysis was conducted in which N2a cells were incubated with an intracellular extract of *Lactobacillus plantarum* CRL2130 and showed that viability was maintained and IL-6 release and reactive oxygen species (ROS) formation, all affected by MPP+, were significantly reduced (In vitro: CRL2130 reduced the oxidative stress and IL-6 (*p* < 0.05); In vivo: CRL2130 improved the motor deficits (*p* < 0.05)).

The study conducted by Xu et al. in 2020 addressed the impact of individual strains of *Lactobacillus* and *Acetobacter* in a *Drosophila* model with PINK mutations, and administered the prebiotic EGCG (eigallocatechin-3-gallate) [104]. The individual strains used were *Lactobacillus plantarum* KJ01, *Acetobacter pomorum* KJ02, *Lactobacillus brevis* KJ03, and *Acetobacter pasteurianus* KJ04. The prebiotic EGCG substantially modified the intestinal microbiota, restoring bacterial abundance, in addition to ameliorating existing dopaminergic deficits. On the other hand, the administration of individual strains of *Lactobacillus* or *Acetobacter* exacerbated the neuronal dysfunction reduced by EGCG (prolonged climbing latency in the PINK1B9 (*p* = 0.013) and PINK1B9+EGCG (*p* = 0.005) flies).

Another recent study was carried out in 2022 by Ilie et al. In this research, an adult zebrafish model (wild-type AB; WT, genetic line) with ROT-rotenone-induced motor impairment was used [105]. Then, the administration of different compounds was included to study their impact on these deficits. The PROBIOTIC group (*Lactobacillus casei* W56, *Lactobacillus acidophilus* W22, *Lactobacillus paracasei* W20, *Lactobacillus salivarius* W24, *Lactobacillus lactis* W19, *Lactobacillus plantarum* W62, *Bifidobacterium lactis* W51, *Bifidobacterium lactis* W52, and *Bifidobacterium bifidum* W23) and LEV+CARB (levodopa and carbidopa) group showed similar neuroprotective effects, decreasing the toxic effects of ROT, favoring neurogenesis and angiogenesis and their maintenance (PROBIO group: positive marking for PCNA, S100b, and GFAP, p53 and cox4i1; ROT+VPA, ROT+LEV/CARB, and ROT+PROBIO: increase in all IHC markers; ROT+PROBIO: PCNA marked a small number of cells).

Evidence suggests that the use of *Lactiplantibacillus plantarum* significantly reduces motor deficits. It also reduces levels of inflammatory factors and maintains neuroprotective effects on dopaminergic neurons, both in vivo and in vitro [99,100,101,102,103,104,105].

#### 3.3.3. Multiple Sclerosis

Research carried out by Mestre et al. in 2020, in an experimental model of MS infected with TMEV-IDD (induced demyelinating disease), analyzed the impact of intervention with Vivomixx (probiotic composed of *Lactobacillus paracasei* DSM 24734, *Lactobacillus plantarum* DSM 24730, *Lactobacillus acidophilus* DSM 24735, *Lactobacillus delbruckeii* subspecies bulgaricus DSM 24734, *Bifidobacterium longum* DSM 24736, *Bifidobacterium infantis* DSM 24737, *Bifidobacterium breve* 24732 and *Streptococcus thermophilus* DSM 24731) during the chronic stage of the disease. It was administered to mice for 70 to 85 days [106].

Consumption of this probiotic improved motor function, showing an increase in horizontal and vertical activity of mice as opposed to the more limited activity of the vehicle-treated mice (*p* < 0.01 vs. TMEV-mice). In addition, it was shown that mice treated with Vivomixx underwent subtle changes in the composition of the gut microbiota. Regarding Vivomixx-TMEV mice group, the relative abundance of *Anaerostipes, Dorea, Oscillospira, Enterobacteraceae* or *Ruminococcus* decreased while *Bacteroides, Odoribacter, Lactobacillus* or *Sutterella* was increased.

This study concluded that the use of a multi-strain probiotic, which includes *Lactobacillus plantarum,* improves motor function [106].

#### 3.3.4. Oxidative Stress

Shang et al. (2022) conducted a study with *Luciobarbus capito* (*L. capito*) in which they investigated the role of Selenium (Se)-enriched *Lactobacillus plantarum* on Cadmium (Cd)-induced oxidative stress [107]. Compared to the control group, the *L. capito* that received the product showed a significant decrease in Cd toxicity after one month of diet (*p* < 0.05), a decrease in oxidative capacity, and improved memory processes.

Xu et al. (2022) conducted a study with a mice model of alcohol-induced cognitive dysfunction. They administered *Lactobacillus plantarum* ST-III (LP-cs) for 28 days [108]. The results showed that LP-cs is able to significantly improve cognitive dysfunction and improved memory and learning. In addition, modifications were found in hippocampal morphology and synaptic dysfunction, reducing the impact of oxidative stress (Control vs. AE and AE vs. AE/LP-cs, * *p* < 0.05, ** *p* < 0.01, *** *p* < 0.001, *n* = 3–6).

The study by Westfall, Lomis and Prakash in 2018 addressed the impact of a probiotic compound (*Lactobacillus plantarum* NCIMB 8826 (Lp8826), *Lactobacillus fermentum* NCIMB 5221 (Lf5221) and *Bifidobacteria longum* spp. *infantis* NCIMB 702,255 (Bi702255)) and a symbiotic compound (probiotic formulation + 0.5% of TFLA power (*Emblica officinalis, Terminalia bellirica* and *Terminalia chebula*)) in an aged male *Drosophila melanogaster* model [109]. The results obtained indicated that both the probiotic compound alone and the symbiotic compound had beneficial effects on metabolic markers associated with aging in this animal model. Specifically, improvements in insulin-like signaling were found, reducing oxidative stress (*p* < 0.01).

Evidence suggests that the use of *Lactiplantibacillus plantarum* improves cognitive function, memory processes and learning, reducing the effect of oxidative stress [107,108,109].

#### 3.3.5. Cognitive Decline and Oxidative Stress

Zaydi et al. (2020) conducted a study in a D-galactose-induced aged rat model [110]. They administered *Lactobacillus plantarum* DR7 for 12 weeks and studied its impact on cognitive impairment and oxidative stress. The results showed that the ingestion of *Lactobacillus plantarum* DR7 improved memory capacity. Anxiety level also was reduced in the DR7-treated group. In addition, the probiotic improved the serotoninergic pathway (MD 5.62; 95% CI 1.78 to 6.11; *p* < 0.05).

This study suggests that the use of *Lactiplantibacillus plantarum* improves cognitive impairment and oxidative stress, memory and anxiety symptomatology [110].

#### 3.3.6. Neurodegenerative Process

Xia et al. (2024) conducted a study in a mice model of D-galactose-induced ageing. They administered *Lactobacillus plantarum* AR113 as a pre-treatment [111]. The results indicated that *Lactobacillus plantarum* AR113 significantly reduced D-galactose-induced oxidative stress injury, reducing cytotoxicity while maintaining cell membrane integrity and enhancing antioxidant enzyme activity. Furthermore, the results suggested that the expression of G protein-coupled receptor 78 (GPR78) and C/EBP homologous protein (CHOP) could be reduced, facilitating the restoration of endoplasmic reticulum (ER) homeostasis, thus activating cellular anti-apoptotic pathways (mRNA expression levels of the marker proteins GPR78 and CHOP were reduced to 0.04 and 0.26 times that of the Mod group, respectively, while the expression of PERK increased to 2.09 times that of the Mod group (*p* < 0.05)).

In conclusion, the use of *Lactiplantibacillus plantarum* reduced the damage caused by oxidative stress associated with the neurodegeneration process [111].

## 4. Discussion

The main objective of this systematic review was to compile all significant findings from human and animal studies investigating the impact of *Lactiplantibacillus plantarum* administration on the improvement of symptomatology in neurodegenerative disease. The results obtained indicated that *Lactiplantibacillus plantarum* either alone or in combination, improves these symptoms.

Regarding PD, motor symptomatology is the diagnostic mainstay of this condition, and the reduction of these symptoms results in an improvement in the patient’s quality of life [70,112,113,114]. Previous studies have shown that the quality of life in PD patients is not only directly related to motor symptoms but also to cognitive impairment, behavioral disturbances, and sleep disorders such as REM sleep disturbance [115,116,117,118,119,120]. The use of probiotics could have positive effects on the quality of life of PD patients by improving motor and non-motor symptomatology.

Human studies showed that *Lactiplantibacillus plantarum*, alone or in combination, improves motor symptoms and increases the proliferation of beneficial bacteria in the gut. These strains have shown neuro-modulatory abilities that delay the progression of PD [92,93]. Similarly, *Lactobacillus plantarum* improved cognitive function in people with MCI, as well as the quality and regularity of sleep [94]. Studies in the same line using different probiotics, such as *Lactobacillus acidophilus*, *Bifidobacterium bifidum*, *Lactobacillus reuteri*, and *Lactobacillus fermentum*, have shown a similar result. Thus, the use of this species produced a reduction in the oxidative stress and showed a neuroprotective effect. Moreover, the patient who took these probiotics showed a better score on the Movement Disorders Society–Unified Parkinson’s Disease Rating Scale. Therefore, the use of probiotics, especially those who belong to the *Lactobacillus* genus, seems to be useful to treat the motor and cognitive symptoms associated with PD.

Regarding the use of probiotics in PD animal models, studies showed a significant improvement in motor deficits following the administration of *Lactiplantibacillus plantarum* alone or in combination. Similarly, it showed neuroprotective effects, reducing dopaminergic neuronal death and glial reactivity, and increasing striatal neurotrophic factors. In addition, neurogenesis and angiogenesis were increased after the probiotic use [99,100,101,102]. Moreover, an in vitro study [103] showed a decrease in ROS. The study performed by Xu et al. in 2020 with a *Drosophila* model showed opposite results indicating that the PROBIO combination exacerbated neuronal disfunction that has been previously alleviated with EGCG (eigallocatechin-3-gallate) [104].

Analyzing other probiotics that could be used in the PD treatment, a recent study using an established *Caenorhabditis elegans* (roundworm) model of synucleinopathy was able to show that worms that received the probiotic strain *Bacillus subtilis* PXN21 hardly formed any alpha-synuclein aggregates, reducing the inflammatory response [121]. In other study conducted by Liu et al. in 2022 in a mouse model of chronic PD, they administered polymannuronic acid (PM) or *Lacticaseibacillus rhamnosus* GG (LGG), or their combination, showing neuroprotective effects independently, and a greater effect when administered in combination. Neurotrophic factor expression was enhanced and the level of Clostridial bacteria was increased. In addition, blood–brain barrier integrity was improved, and apoptosis in the striatum was inhibited [122]. It is therefore necessary to identify other strains that can be administered alone or in combination to enhance the neuroprotective effects.

Analyzing the results obtained from studies addressing AD, the use of *Lactiplantibacillus plantarum* alone or in combination showed benefits. Specifically, it improved AD-associated memory function [96], and neurodegeneration was reduced through processes in the brain’s ATPase system as well as the neuroprotective role exerted by the administered probiotic [97]. In addition, brain glucose metabolism, involved in the degenerative processes associated with AD, was improved [95]. Moreover, neuronal survival and synapse function were also protected. Finally, modulation of the gut microbiota led to a modification of the microbial profile, which exerts a neuroprotective function [95].

On the other hand, the study focused on gender differences, showing that *Lactiplantibacillus plantarum* improves memory in female mice and reduced TNF-a levels, although these improvements were not found in male mice [98].

In this line, other probiotics that have shown positive effects in ameliorating the symptomatology of Alzheimer’s disease (AD) include *Bifidobacterium longum*, which improved memory and plasticity in rats. Interventions with *Lactobacillus acidophilus*, *Lactobacillus fermentum*, *Bifidobacterium lactis*, and *Bifidobacterium longum* have demonstrated positive effects in both human and animal models [97,123,124].

The study of the impact of *Lactiplantibacillus plantarum* in MS intervention showed a significant improvement in the motor symptoms associated with the pathology, as well as an increase in anti-inflammatory activity in microglia, reducing oxidative stress. Furthermore, in the study of cognitive function, the administration of *Lactiplantibacillus plantarum* showed a decrease in memory loss [106].

Regarding the use of other probiotics in the MS treatment, others *Lactobacillus* spp., *Bifidobacteriums* (such as *B. animalis*) and *Streptococcus* showed a modulation in the CNS inflammation by T helper type 1 (Th1) and Th17 cells, which produce pro-inflammatory cytokines that damage the blood–brain barrier [125,126,127,128].

Moreover, studies analyzing oxidative stress, cognitive impairment, and neurodegenerative processes, along with the impact of various strains of *Lactiplantibacillus plantarum*, have shown significant improvements in reducing the loss of mitochondrial integrity; improvements in cognitive processes such as learning and memory, an increase in the activity of antioxidant enzymes, and a reduction in metabolic stress and inflammation [129,130,131,132,133]. As in other studies, the administration of *Lactiplantibacillus plantarum* was shown to modify hippocampal morphology and improve synaptic dysfunction [124,134,135].

While the utilization of probiotics appears promising in the treatment of neurodegenerative diseases, it is important to note that it is not the sole option available. An alternative approach involves the modulation of gut microbiota through fecal transplantation from healthy individuals, which has demonstrated efficacy in restoring eubiosis. This restoration enhances anti-inflammatory mechanisms and mitigates cellular oxidative processes. Notably, the application of this technique has proven to be a valuable treatment for a spectrum of neurodegenerative conditions, including PD, AD, MS, and neurodegenerative processes associated with oxidative stress [136,137,138,139].

### Limitations and Future Research

The main limitation of this systematic review was the lack of clinical studies, which is consistent with the novelty of this area of study. However, this review is relevant and significant, as it provides the latest information currently available, focusing on the probiotic interventions. The inclusion of these details enables us to identify starting points for future research with a solid and rigorous empirical foundation.

Additional limitations arise from the heterogeneity of the utilized probiotics, as well as variations in quantities of CFUs and combinations of strains within probiotic formulations. Moreover, the mixture of probiotics with other substances further contributes to the complexity of these limitations and makes it difficult to reach solid conclusions on the beneficial effects of *Lactiplantibacillus plantarum* use alone or in combination. It is essential to emphasize that further research is necessary to elucidate the mechanisms of action for each probiotic. Additionally, determining the most appropriate administration protocol in each case is crucial to ensure the neuroprotective effects of the probiotic under consideration.

There is also a reduced number of studies indicating non-beneficial or non-existent effects of probiotics administration and it is necessary to also have information about these cases, to improve interventions.

This systematic review is the first to report and highlight the effects of *Lactiplantibacillus (Lactobacillus) plantarum*, alone or in combination, on the neurodegeneration processes present in neurodegenerative diseases. This information supports the development of interventions incorporating this probiotic to alleviate motor, cognitive, and behavioral symptoms linked to these pathologies.

## Figures and Tables

**Figure 1 ijms-25-03010-f001:**
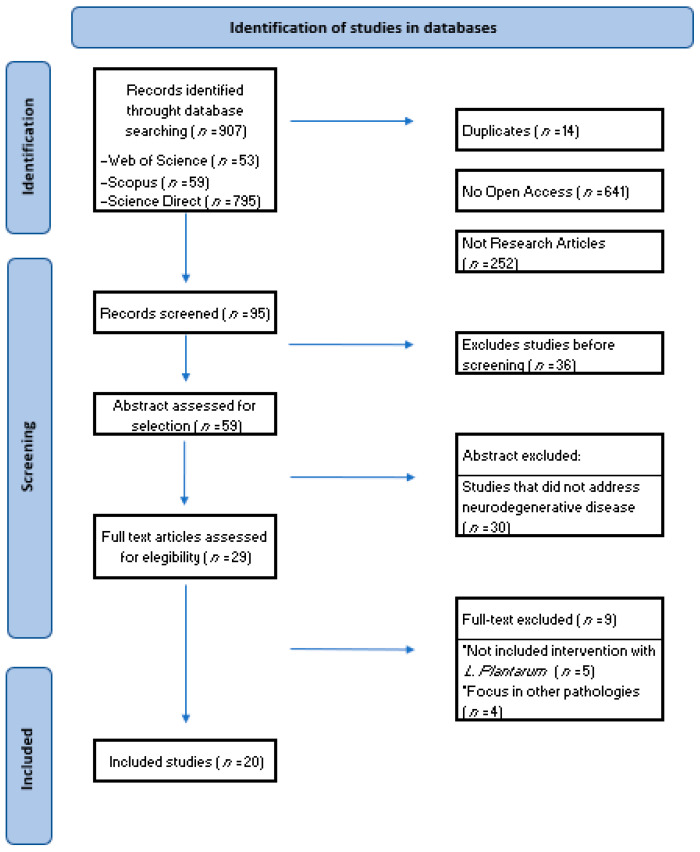
Flow diagram of the search and selection process. Twenty (*n* = 20) studies met the eligibility criteria and investigated the effects of *Lactiplantibacillus plantarum* or *Lactobacillus plantarum* alone or in combination on neurodegenerative processes (3 studies in humans and 17 studies in animal models).

**Table 1 ijms-25-03010-t001:** Effects of *Lactiplantibacillus plantarum*, alone or in combination with neurodegenerative diseases (in human and animal model).

	Probiotics	Population	Methodology	Intervention	Results	References	
	Parkinson’s Disease						
HUMAN MODEL	*Lactobacillus acidophilus* NCIMB 30175, *Lactobacillus plantarum* NCIMB 30173, *Lactobacillus rhamnosus* NCIMB 30174 and *Enterococcus faecium* NCIMB 30176	Fecal samples from six donors (three healthy control subjects, three diagnosed with PD)	In vitro dynamic, multi-compartment gastrointestinal model	Fermentation with probiotic for 48 h	Increased levels of Firmicutes, Actinobacteria and Bacteroidetes in PD patients. Modulation of the intestinal microbiota that delays pathology progression	10.1016/j.ijpx.2021.100087	[92]
	*Lactobacillus plantarum* PS128	25 patients	Open-label, single-arm, baseline-controlled trial	PS128 (2 capsules containing 30 billion colony-forming units per capsule) daily for 12 weeks	Improved motor scores and quality of life	10.3389/fnut.2021.650053	[93]
	MCI (Mild Cognitive Impairment)						
	*Lactobacillus plantarum* BioF-228, *Lactococcus lactis* BioF-224, *Bifidobacterium lactis* CP-9, *Lactobacillus rhamnosus* Bv-77, *Lactobacillus johnsonii* MH-68, *Lactobacillus paracasei* MP137, *Lactobacillus salivarius* AP-32, *Lactobacillus acidophilus* TYCA06, *Lactococcus lactis* LY-66, *Bifidobacterium lactis* HNO19, *Lactobacillus rhamnosus* HNO01, *Lactobacillus paracasei* GL-156, *Bifidobacterium animalis* BB-115, *Lactobacillus casei* CS-773, *Lactobacillus reuteri* TSR332, *Lactobacillus fermentum* TSF331, *Bifidobacterium infantis* BLI-02, *and Lactobacillus plantarum* CN2018	42 patients	Pilot randomized controlled trial (RCT)	2 g (>2 × 10^10^ CFU/g) probiotics daily for 12 weeks	Cognitive function and sleep quality were improved	10.1016/j.gerinurse.2023.03.006	[94]
	Alzheimer’s Disease						
ANIMAL MODEL	SLAB51: *Streptococcus thermophilus* DSM 32245, *Bifidobacterium lactis* DSM 32246, *B. lactis* DSM 32247, *Lactobacillus acidophilus* DSM 32241, *Lactobacillus helveticus* DSM 32242, *Lactobacillus paracasei* DSM 32243, *Lactobacillus plantarum* DSM 32244 and *Lactobacillus brevis* DSM 27961	48 AD male mice triple-transgenic, B6; 129-Psen1^tm1Mpm^ Tg (amyloid precursor protein [APP] Swe, tauP301L) 1Lfa/J (named 3xTg-AD) and the wt B6129SF2 mice	Treatment with probiotic vs. Sham	SLAB51 (200 billion bacteria/kg/d) daily for 16 and 48 weeks	It improved impaired brain glucose metabolism implicated in AD pathogenesis, delaying disease progression	10.1016/j.neurobiolaging.2019.11.004	[95]
	NK151: *Lactobacillus plantarum*; NK173: *Bifidobacterium longum*	35 specific pathogen-free C57BL/6 mice	Treatment with probiotic vs. Sham	*Escherichiacoli* K1 (EC, 1 × 10^9^ CFU per mouse per day) were orally gavage daily for 5 days	Reduced memory impairment and modulated gut microbial profile	10.1039/d1fo02167b	[96]
	*Lactobacillus plantarum* MTCC1325	48 healthy adult male Wistar rats	The hippocampal (HP) and cerebral cortex (CC) brain regions of each animal were isolated	*Lactobacillus plantarum* MTCC1325 daily for 60 days	Neurodegeneration was reduced by the protective effect on the ATPase system in the brain	10.15171/bi.2016.27	[97]
	VSL#3 ^®^: *Lactobacillus plantarum*, *Lactobacillus delbrueckii* subsp. *Bulgaricus*, *Lactobacillus paracasei*, *Lactobacillus acidophilus*, *Bifidobacterium breve*, *Bifidobacterium longum*, *Bifidobacterium infantis* and *Streptococcus salivarius* subsp. *Thermophilus*	140 mice: 70 male and female AppNL-G-F (AD) and 70 male and female C57BL/6J (wild type, WT)	Treatment with probiotic vs. Sham vs. ABX (antibiotic) vs. ABX+probiotic	VSL#3 ^®^ (4 × 10^9^ UFC/día/mice) for 8 weeks	All groups of female mice with intervention improved mnesic ability but none affected memory in male mice; reduced brain levels of TNF-α in female AppNL-G-F mice	10.3390/cells10092370	[98]
	Parkinson’s Disease						
	*Lactobacillus plantarum* PS128	90 six-week-old male C57BL/6J mice	Treatment with probiotic vs. Sham	PS128 (10^9^ CFU in 200 μL saline) daily for 28 days	Motor deficits were improved and dopaminergic neuronal death was alleviated. Glial reactivity is reduced and striatal neurotrophic factors are increased	10.1016/j.bbi.2020.07.036	[99]
	*Lactobacillus plantarum* PS128	40 eight-week-old male C57BL/6J mice	Treatment with probiotic vs. Sham	PS128 (10^9^ CFU/d) (week 1 to 6) and rotenone	Modified the microbial profile and maintained neuroprotective effects by increasing the expression of the suppressor of cytokine signaling	10.3390/ijms24076794	[100]
	*Lactobacillus plantarum* DP189	90 male C57BL/6 mice	Treatment with probiotic vs. Sham	DP189 (0.2 mL) daily for 14 days	Improved motor symptoms and increased levels of monoamines in the nervous system. In addition, it promoted neuronal survival	10.1016/j.jff.2021.104635	[101]
	*Lactobacillus plantarum* CRL 2130, *Streptococcus thermophilus* CRL 808, and *Streptococcus thermophilus* CRL 807	56 eight-week-old male C57BL/6 mice	Treatment with probiotic vs. Sham	*L. plantarum* CRL 2130, *S. thermophilus* CRL 808, *S. thermophilus* CRL 807 or the bacterial mixture (100 µL that contain 8 ± 2 × 10^8^ CFU/mL of each strain, individually or as a mixture daily) for 4 days	Both alone or in combination, the strains produced significantly improved motor skills, with possible involvement of vitamin B interaction, and protective immune response to organ inflammation	10.1016/j.nut.2020.110995	[102]
	*Lactobacillus plantarum* CRL2130, *Lactobacillus plantarum* CRL725	35 neuro-2a (N2a) neuroblastoma cell line (ATCC CCL131) mice	In vitro: Half of each brain was removed for the determination of cytokines; The other half of the brain was stored in 10% paraformaldehyde/PBS during 24 h and then embedded in paraffin; In vivo: Treatment with probiotic vs. Sham	In vitro: N2a cells were incubated with intracellular extract from *L. plantarum* CRL2130. In vivo: *Lactobacillus plantarum* CRL2130, *Lactobacillus plantarum* CRL725 (100 μL) once	In vitro: cells-maintained viability and significantly decreased IL-6 release and reactive oxygen species (ROS) formation. In vivo: attenuated motor deficits and prevented dopaminergic neuronal death	10.1007/s11064-021-03520-w	[103]
	*Lactobacillus plantarum* KJ01, *Acetobacter pomorum* KJ02, *Lactobacillus brevis* KJ03, and *Acetobacter pasteurianus* KJ04	30 to 40 adult *Drosophila*	Treatment with probiotic vs. EGCG	*Lactobacillus plantarum* o *Acetobacter pomorum* (5 g food/1 mL of bacterial suspension) for 20 days (OD_600_ = 1, approximately 5 × 10^7^ viable cells/mL)	LP KJ01 exacerbated PD symptoms and eliminated EGCG (eigallocatechin-3-gallate)-mediated symptom enhancement. In addition, it aggravated neuronal loss and hindered EGCG-mediated enhancement.	10.1096/fj.201903125RR	[104]
	PROBIO: *Lactobacillus casei* W56, *Lactobacillus acidophilus* W22, *Lactobacillus paracasei* W20, *Lactobacillus salivarius* W24, Lactobacillus lactis W19, *Lactobacillus plantarum* W62, *Bifidobacterium lactis* W51, *Bifidobacterium lactis* W52, and *Bifidobacterium bifidum* W23	120 adult (6–8 months) wild-type AB (WT, genetic line) zebrafish	Treatment with probiotic vs. Sham	PROBIO (3 g × 28 envelopes) were dissolved directly into 100 mL distilled water as a unique dose per day	ROT triggers apoptosis and increases PARKIN and PINK1 expression. LEV/CARB and PROBIO maintain neurogenesis and angiogenesis, exerting neuroprotective functions and decreasing the impact of ROT	10.3390/antiox11102040	[105]
	Multiple Sclerosis						
	*Lactobacillus paracasei* DSM 24734, *Lactobacillus plantarum* DSM 24730, *Lactobacillus acidophilus* DSM 24735, *Lactobacillus delbruckeii* subspecies *bulgaricus* DSM 24734, *Bifidobacterium longum* DSM 24736, *Bifidobacterium infantis* DSM 24737, *Bifidobacterium breve* 24732 and *Streptococcus thermophilus* DSM 24731	Fecal samples freshly obtained from each mouse on day 70 and 85	Treatment with probiotic vs. Sham	2 times a week: 3 × 10^8^ CFU (100 μL) of Vivomixx for 70 to 85 days	Motor function was improved. Increased anti-inflammatory activation in the microglia	10.1080/19490976.2020.1813532	[106]
	Oxidative Stress						
	Se-enriched *Lactobacillus plantarum* CCFM8610	270 Luciobarbus capito	Treatment with probiotic vs. Sham	Diet of 1–2% body weight twice a day for 30 days	Reversed Cd toxicity in the blood and brain, improving antioxidant capacity and reducing memory loss	10.1016/j.ecoenv.2022.113890	[107]
	*Lactobacillus plantarum* ST-III AB161 (CGMCC 22782)	24 male ICR mice	Treatment with probiotic vs. Sham	Free access for 28 days	Significantly improved cognitive dysfunction (memory and learning). Modifications in hippocampal morphology and synaptic dysfunction were found, reducing the impact of oxidative stress	10.3389/fnins.2022.976358	[108]
	*Lactobacillus plantarum* NCIMB 8826 (Lp8826), *Lactobacillus fermentum* NCIMB 5221 (Lf5221) and *Bifidobacteria longum* spp. infantis NCIMB 702255 (Bi702255)	Wildtype *Drosophila melanogaster* (Oregon R)	Treatment with probiotic vs. synbiotic	Probiotic formulation: 3.0 × 10^9^ CFU/mL of probiotics with equal distribution between Lp8826 (1.0 × 10^9^ CFU/mL), Lf5221 (1.0 × 10^9^ CFU/mL) and Bi702255 (1.0 × 10^9^ CFU/mL). Synbiotic formulation: probiotic formulation + 0.5% of TFLA powder	The probiotic and synbiotic reduced metabolic stress levels and inflammation, improved oxidative stress and loss of mitochondrial integrity. Synbiotic formulation maintains results better than separate formulation	10.1038/s41598-018-25382-z	[109]
	Cognitive decline and oxidative stress						
	*Lactobacillus plantarum* DR7	6 Male Sprague Dawley rats	Treatment with probiotic vs. Sham	DR7 (10^9^ CFU/day) for 12 weeks	Memory capacity improved. Anxiety also improved in the DR7-treated group. In addition, protective effects on serotonergic pathways were demonstrated	10.3920/BM2019.0200	[110]
	Neurodegenerative process						
	*Lactobacillus plantarum* AR113	32 male C57BL/6J mice	Treatment with probiotic vs. Sham	AR113 (1 × 10^9^ CFU/mL) daily	It significantly reduced oxidative stress injury induced by D-galactose, decreasing its cytotoxicity. In addition, cell membrane integrity was maintained and antioxidant enzyme activity was increased	10.26599/fshw.2022.9250076	[111]

## Data Availability

Not applicable.

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
