# Peer review of "Lactiplantibacillus (Lactobacillus) plantarum as a Complementary Treatment to Improve Symptomatology in Neurodegenerative Disease: A Systematic Review of Open Access Literature"

_ijms, 2024, doi:10.3390/ijms25053010_

Round 1

Reviewer 1 Report

Comments and Suggestions for Authors

This is a study addressing the use of Lactobacillus plantarum by mean of a systematic review. However, the paper does not present a systematic evaluation but just a narrative review. A systematic review should use established methods, not only for the choice of the articles but also a strict statistical analysis of the studies chosen in order to examine evidence for the intervention. In order to do so the precise outcome should be defined. However, the strong confusion the paper makes a between symptoms and biological processes impairs such an evidence-seeking analysis.

The abstract should be better structured around the actual study, no so much on the background. Some addition words about methodology are needed and a conclusion should be presented.

The introduction should be very much shortened, and focused on the aim of the study.

There are a lot of propaedeutic statements with are not needed in a scientific article. For example, the description of the core clinical and pathological aspects of the three disease is not useful for the study.

Care should be also taken in the suggestion, that dysbiosis is the cause of neurodegeneration, as it could as well be the consequence of the disease. Throughout the introduction a better care should be give to carefully differentiate potentially neuroprotective and symptomatic effects.

Flaws are present in the method section:

It is not clear

The inclusion criteria are not clear. The first criterion will by itself cause a bias since only positive studies would be taken, the ones which shaw a direct relationship between intervention and changes in symptomatology. The second stresses on the pathophysiological mechanism, but this can mostly not be assessed in clinical trials. The third makes a confusion between truly degenerative disorders (PD and AD), mixed pathogenesis (MS), mechanisms (oxidative stress), purely clinical construct (MCI) and other (without definition).

 The result section continues on the confusion between symptoms and disease processes. There are a large number of generalizing statements without proper description and quotation.  

Comments on the Quality of English Language

There are numerous minor points around English language, only a few of them being mentioned here:

Some statements are inexact or vage. For examples

-Line 10 there is presently no treatment to reduce neuronal death in order to change the onset.

-Line 28: the evidence for the 2050 projection is not given (no paper quoted)

-line 33: what is traditionally studied?

-line 59: MS is both a neuroinflammatory and a neurodegenerative condition

-line 72: permanence during the synapse: here a mixture of anatomy and time. Acetylcholine should not be permanently present in the synapse but only transiently

-line 125: 2024 papers will mostly not have been published before November 2023

-line 134: processes do not have symptoms, diseases may

Etc etc

There are also flaws in style, which should be corrected, for examples:

-line 40: all neurodegenerative and neurodegeneration: unnecessary repetition.

-line 41: repetition after prior sentence (no need to repeat that AD is a neurodegenerative condition.

Etc etc

Author Response

Dear reviewer:

Following the indications of Reviewer #2 we have performed a new search. This is due to the change of nomenclature of some phyla and bacterial species, which explains the change of title, methodology (updated) and results. Four new articles containing this new nomenclature have been included. This is also explained in the Introduction.

Reviewer #1

Comment: “This is a study addressing the use of Lactobacillus plantarum by mean of a systematic review. However, the paper does not present a systematic evaluation but just a narrative review. A systematic review should use established methods, not only for the choice of the articles but also a strict statistical analysis of the studies chosen in order to examine evidence for the intervention. In order to do so the precise outcome should be defined. However, the strong confusion the paper makes a between symptoms and biological processes impairs such an evidence-seeking analysis.”

Response: This study is a systematic review of the literature, and there is no associated meta-analysis, so statistical analyses have not been performed because they are not necessary in a systematic review. However, we found your suggestion very interesting and have therefore included the statistics for each study in the Results section.

Comment:The abstract should be better structured around the actual study, no so much on the background. Some addition words about methodology are needed and a conclusion should be presented.”

Response: Thank you for your suggestion. We have modified the Abstract according to your instructions.

“This systematic review addresses the use of Lactiplantibacillus plantarum in the symptomatological intervention of neurodegenerative disease. The existence of gut microbiota dysbiosis has been associated with systemic inflammatory processes present in neurodegenerative disease, creating the opportunity of new strategies for their treatment. This involves modifying the strains that constitute the gut microbiota to enhance synaptic function through the gut-brain axis. Recent studies have evaluated the beneficial effects of the use of Lactiplantibacillus plantarum on motor and cognitive symptomatology, alone and in combination with other bacteria. This systematic review includes 20 research articles (n = 3 in human and n = 17 in animal models). The main result of this research was that the use of Lactiplantibacillus plantarum alone or in combination produced improvement in symptomatology related to neurodegenerative disease. Nevertheless, one of the studies included reported negative results administration of Lactiplantibacillus plantarum. This systematic review provides current and relevant information about the use of this probiotic in pathologies that present neurodegenerative processes such as Alzheimer's disease, Parkinson's disease and Multiple Sclerosis”.

Comment:The introduction should be very much shortened, and focused on the aim of the study. There are a lot of propaedeutic statements with are not needed in a scientific article. For example, the description of the core clinical and pathological aspects of the three disease is not useful for the study. Care should be also taken in the suggestion, that dysbiosis is the cause of neurodegeneration, as it could as well be the consequence of the disease. Throughout the introduction a better care should be give to carefully differentiate potentially neuroprotective and symptomatic effects.”

Response: Following your suggestion, we have eliminated certain parts of the Introduction that were not focused on the object of this research.

We have added the explanatory note that it is not possible to determine whether dysbiosis is a cause or a consequence. Moreover, when it has been stated in the paper that dysbiosis is a cause of certain pathology due, among others, to the increased intestinal permeability associated with dysbiosis, which would increase neuronal damage; or the increased of toxin levels like LPSs and other mechanism like monocyte infiltration. For example: 10.1159/000518147; 10.3390/ijms24044047; 10.1016/j.tins.2013.01.005; 10.1089/jmf.2014.7000; 10.3390/biomedicines10020289.

 Comment:The inclusion criteria are not clear. The first criterion will by itself cause a bias since only positive studies would be taken, the ones which shaw a direct relationship between intervention and changes in symptomatology. The second stresses on the pathophysiological mechanism, but this can mostly not be assessed in clinical trials. The third makes a confusion between truly degenerative disorders (PD and AD), mixed pathogenesis (MS), mechanisms (oxidative stress), purely clinical construct (MCI) and other (without definition)”.

Response: The first inclusion criteria does not state that the observed changes have to be positive. In fact, with the new search proposed by reviewer 2, a study showing a negative impact of Lactiplantibacillus plantarum use on motor symptomatology in PD is included (10.1096/fj.201903125RR). The second inclusion criterion does not emphasize the pathophysiological mechanism, but refers to Lactiplantibacillus plantarum being used as a therapeutic strategy in neurodegenerative processes. The third inclusion criterion includes the most frequent pathologies that cause neuronal death, such as neurodegenerative diseases (AD, PD, MCI) or MS (autoimmune and demyelinating pathology, which causes neuronal death), and oxidative stress has been included because it is a common factor in all of them. This is detailed in the Introduction, as well as the differences in the etiology of each of them. However, it has been modified for a better understanding.

“That the studies included any of the pathologies studied in this systematic review, which involve neuronal death associated to AD, PD, MCI and MS, oxidative stress (and other neurodegenerative processes”.

 Comment:The result section continues on the confusion between symptoms and disease processes. There are a large number of generalizing statements without proper description and quotation”.  

 Response: Thank you for the suggestion. We have reviewed the Results section. The statistics of all included studies have been added. In terms of wording, these results have been described according to the articles reviewed.

Comment: Some statements are inexact or vage. For examples:

 -Line 10 there is presently no treatment to reduce neuronal death in order to change the onset.

Response: We have modified the Abstract following your comment.

-Line 28: the evidence for the 2050 projection is not given (no paper quoted)

Response: Line 31:5. Nichols E, Steinmetz JD, Vollset SE, Fukutaki K, Chalek J, Abd-Allah F, et al. Estimation of the global prevalence of dementia in 2019 and forecasted prevalence in 2050: an analysis for the Global Burden of Disease Study 2019. Lancet Public Health. 2022 Feb;7(2): e105–25”.

-Line 33: what is traditionally studied?

 Response: Line 34: There have been articles since at least 1986 on the effect of oxidative stress on neurodegenerative processes. Therefore, since then a great deal of evidence has been created while the study of the impact of dysbiosis on these processes is much more recent (10.1111/j.1471-4159.1986.tb00615.x).

-Line 59: MS is both a neuroinflammatory and a neurodegenerative condition.

 Response: Line 56: As the empirical evidence explains, DM in an autoimmune disease. For example:

Multiple sclerosis (MS) is a heterogeneous, chronic, non-traumatic, disabling, autoimmune disease characterized by inflammation, demyelination, oligodendropathy, astrogliosis, neuronal and axonal degeneration, most commonly affecting young adults, usually above the second and below the fourth decade of life, with both white and gray matter involvement” (40), other evidences: (Anderson et al., 1992, Trapp and Nave, 2008, Bo et al., 2006).

Multiple sclerosis (MS) is an immune-mediated demyelinating disease of the central nervous system (CNS) with cerebral neurodegeneration” (41).

  1. Adiele RC, Adiele CA. Metabolic defects in multiple sclerosis. Mitochondrion. 2019 Jan; 44:7–14.
  2. Lo Sasso B, Agnello L, Bivona G, Bellia C, Ciaccio M. Cerebrospinal Fluid Analysis in Multiple Sclerosis Diagnosis: An Update. Medicina (B Aires). 2019 Jun 4;55(6):245.

-Line 72: permanence during the synapse: here a mixture of anatomy and time. Acetylcholine should not be permanently present in the synapse but only transiently

Response: Following previous comment we have eliminated different paragraphs of Introduction and this information has been removed.

-Line 125: 2024 papers will mostly not have been published before November 2023

Response: The publication dates recorded for each article are those established by the journal. This can be confirmed in each of the publications included in this study.

-Line 134: processes do not have symptoms, diseases may

Response: Line 126: Following your comments, this aspect has been modified for a better understanding.

“In both human and animal model studies, the inclusion criteria required: i) that there was a direct relationship between the use of Lactobacillus plantarum OR Lactiplantibacillus plantarum and changes in the symptomatology of neurodegenerative disease”

There are also flaws in style, which should be corrected, for examples:

-line 40: all neurodegenerative and neurodegeneration: unnecessary repetition.

Response: Line 43: Following your comments, we have modified this stile mistake.

“More than 100 neurodegenerative diseases or conditions associated with neuronal death have been described”.

-line 41: Line 45: repetition after prior sentence (no need to repeat that AD is a neurodegenerative condition.

 Response: Following your comments, we have modified this stile mistake.

“AD is the most common form of dementia among older adults, with a prevalence of more than 46 million people affected worldwide (20,21)”

Reviewer 2 Report

Comments and Suggestions for Authors

The manuscript of a review paper by Ana Isabel Beltrán-Velasco et al. is a valuable and rather comprehensive overview of using Lactiplantibacillus (Lactobacillus) plantarum to help to improve neurodegenerative disease symptoms.

The gut-brain axis has received immense

attention due to the opportunities to improve human health and provide additional support to relieve various conditions and diseases. The review manuscript gives a systematic overview of the recent advances in probiotic treatment to alleviate neurodegenerative processes such as Alzheimer's disease, Parkinson's disease and Multiple Sclerosis.

There are some comments:

Throughout the text. From 2020, the correct species name is Lactiplantibacillus plantarum. The official name and the synonym should appear both in the abstract and manuscript body. https://lpsn.dsmz.de/species/lactiplantibacillus-plantarum Former Lactobacillus genus was divided into more than 20 new genera.

Lines 83-85, 94-95, 172-173. The current systematic names of the bacterial phyla have been changed. Please review from the official nomenclature: https://lpsn.dsmz.de/domain

Lines 106-107, 346-347 and elsewhere. Please check the correct systematic names of the mentioned species.

Lines 108-109. It should be family Lactobacillaceae.

Line 126. Are the authors certain that no significant studies were left out as only the former synonym of the bacterium species but not the current one (Lactiplantibacillus plantarum) as search terms? Confirmation of the search results should be provided.

The table presented contains several formatting errors. Where possible the dose of probiotics should be indicated in 10 exponents.

Author Response

Reviewer #2

Comment: “The manuscript of a review paper by Ana Isabel Beltrán-Velasco et al. is a valuable and rather comprehensive overview of using Lactiplantibacillus (Lactobacillus) plantarum to help to improve neurodegenerative disease symptoms. The gut-brain axis has received immense attention due to the opportunities to improve human health and provide additional support to relieve various conditions and diseases. The review manuscript gives a systematic overview of the recent advances in probiotic treatment to alleviate neurodegenerative processes such as Alzheimer's disease, Parkinson's disease and Multiple Sclerosis.”

Response: Many thanks for your positive comments.

Comment: “Throughout the text. From 2020, the correct species name is Lactiplantibacillus plantarum. The official name and the synonym should appear both in the abstract and manuscript body. https://lpsn.dsmz.de/species/lactiplantibacillus-plantarum Former Lactobacillus genus was divided into more than 20 new genera”.

 Response: Thank you very much for your comment. Initially, we used the previous name because there was a greater number of articles associated with this name. However, thanks to your comment, we have conducted a new search that includes both types of names. Following this search, a total of 4 new articles have been included in the review. Your comment has allowed us to enhance the quality of the presented review, and for that, we are very grateful.

Comment: “Lines 83-85, 94-95, 172-173. The current systematic names of the bacterial phyla have been changed. Please review from the official nomenclature: https://lpsn.dsmz.de/domain. Lines 106-107, 346-347 and elsewhere. Please check the correct systematic names of the mentioned species. Lines 108-109. It should be family Lactobacillaceae. Line 126. Are the authors certain that no significant studies were left out as only the former synonym of the bacterium species but not the current one (Lactiplantibacillus plantarum) as search terms? Confirmation of the search results should be provided.”

Response: Thank you very much for the suggestion and for the help with the website provided. We have reviewed all the phyla and species and have updated them. However, we would like to note that the names of the bacteria have been preserved in the results and discussion sections as specified in the articles included.

Comment: “The table presented contains several formatting errors. Where possible the dose of probiotics should be indicated in 10 exponents.”

Response: We have revised the table and corrected the formatting errors observed.

Round 2

Reviewer 2 Report

Comments and Suggestions for Authors

The comments have been suffciently addressed and the manuscript improved.

Comments on the Quality of English Language

The language is suitable. 

Author Response

Dear Reviewer,

Thank you for your positive comments during the process.

We have reviewed and changed the minor errors in the English language.

Best regards,